# Long-Term Lower Limb Motor Function Correlates with Middle Cerebellar Peduncle Structural Integrity in Sub-Acute Stroke: A ROI-Based MRI Cohort Study

**DOI:** 10.3390/brainsci13030412

**Published:** 2023-02-27

**Authors:** Daming Wang, Lingyan Wang, Dazhi Guo, Shuyi Pan, Lin Mao, Yifan Zhao, Liliang Zou, Ying Zhao, Aiqun Shi, Zuobing Chen

**Affiliations:** 1Departments of Physical Medicine and Rehabilitation, The First Affiliated Hospital, Zhejiang University School of Medicine, Hangzhou 310003, China; 2Departments of Physical Medicine and Rehabilitation, Zhejiang Chinese Medical University Affiliated Jinhua TCM Hospital, Jinhua 321000, China; 3Department of Hyperbaric Oxygen, The Sixth Medical Center of PLA General Hospital, Beijing 100142, China

**Keywords:** cerebral infarction, diffusion tensor imaging, prognosis, middle cerebellar peduncle, walking ability

## Abstract

Crossed cerebellar diaschisis (CCD) has been widely investigated in patients with supratentorial stroke. However, the role of CCD in lower limb recovery after stroke is still unknown. In this study, using a region-of-interest-based analysis of diffusion tensor imaging (DTI), a total of 44 cases of stroke within 3 months onset were enrolled for assessment of the cerebral peduncle (CP) and middle cerebellar peduncles (MCP) in CCD. Compared with the control group, the fractional anisotropy ratio (rFA) and laterality index (LI) of the CP and MCP in the stroke group significantly decreased. The rFA of the MCP (unaffected side/affected side) showed a more significant correlation with 1-year paresis grading (PG), lower extremity PG, upper extremity PG, National Institutes of Health Stroke Scale (NIHSS), and functional independence measure (FIM) motor item score, in comparison to the rFA of the CP (affected side/unaffected side) (r = −0.698 vs. r = −0.541, r = −0.651 vs. r = −0.386, r = −0.642 vs. r = −0.565, r = −0.519 vs. r = −0.403, and r = 0.487 vs. r = 0.435, respectively). Furthermore, the LI of the CP had a more significant association with 1-year Brunel Balance Assessment (BBA), upper extremity PG, and Modified Rankin Scale (mRS) as compared to the LI of the MCP (r = 0.573 vs. r = 0.452; r = −0.554 vs. r = −0.528; and r = −0.494 vs. r = −0.344, respectively). We set the cutoff point for the MCP rFA at 0.925 (sensitivity: 79% and specificity: 100%) for predicting lower extremity motor function prognosis and found the receiver operating characteristic (ROC) curve of MCP rFA was larger than that of CP rFA (0.893 vs. 0.737). These results reveal that the MCP may play a significant role in the recovery of walking ability after stroke.

## 1. Introduction

Crossed cerebellar diaschisis (CCD) is very common in all phases of stroke, occurring in 15.6–46.2% of cases [1]. Positron emission tomography (PET) can show reversible hypoperfusion in the cerebellum 3 h after stroke and contralateral to the lesion, while diffusion tensor imaging (DTI) can measure CCD in the subacute and chronic phases [2]. Both PET and DTI studies have shown that the occurrence and extent of CCD are associated with poor overall prognosis in stroke patients. However, there is no evidence to show that CCD is related to motor recovery, Walking ability is relatively easy to recover after sub-acute stroke and occurs before upper limb or hand function in sub-acute stroke [3,4].

Studies of Alzheimer’s disease, Parkinson’s disease and aphasia have confirmed that the cerebellum may be involved in modifying the performance of learned sequences such as cycling that are based on procedural memory-related brain activity (cerebellum–striatum–frontal circuit) [5,6,7,8].

The medial temporal lobe is responsible for declarative memory-related brain activity that deals mainly with the association and integration of newly acquired motor memory or skills. The difference in movement mode between walking ability and fine motor function of upper limbs parallels the difference between programmed movement and technical activities. The cerebellum is responsible for programmed memory, and together with its neural pathways connected to the brain, such as CCD phenomenon, it may be linked to the poor recovery of walking after stroke. It is, therefore, reasonable to speculate that secondary degeneration of CCD in the middle cerebellar peduncles (MCP), the main pathway through which the frontal lobe enters the cerebellum, may be associated with different long-term lower extremity motor ability in patients with hemiplegia following stroke [9,10]

The aim of this study was to prospectively assess hemiplegic patients within 3 months of first supratentorial stroke, for fractional anisotropy (FA) values of cerebral peduncle (CP) and MCP. In addition, we investigated the relationship between FA values and the recovery of lower extremity motor ability recovery in sub-acute stroke, compared with upper extremity motor ability.

## 2. Materials and Methods

### 2.1. Subjects

We consecutively enrolled all subacute stroke patients who were transferred to the Department of Rehabilitation Medicine from February 2013 to June 2016. The inclusion criteria were as follows: (1) diagnosis of the first onset of cerebral infarction within the middle cerebral artery region and/or supratentorial intracranial hemorrhage (sICH) by computed tomography or magnetic resonance imaging; (2) patients between 18 and 75 years old; (3) patients admitted within 14 days to 3 months after stroke onset; (4) patients providing their informed consent; and (5) DTI was performed at least 2 weeks after the onset of hemiparesis. We excluded patients with any of the following: (1) history of previous stroke; (2) history of traumatic brain damage or brain tumor; (3) presence of neuromuscular disease (e.g., amyotrophic lateral sclerosis and myasthenia gravis); (4) unstable medical condition disabling completion of the clinical trial; (5) no metal implants in head, face and heart; and (6) fluid-attenuated inversion recovery sequences in each patient showed no imaging abnormalities in the cerebellum or cerebral hemisphere contralateral to the lesion, which ruled out previous structural brain damage. All patients received more than 20 routine rehabilitation treatments and follow-up for 6–12 months. This study was approved by institutional review board of The First Affiliated Hospital, Zhejiang University School of Medicine, Hangzhou, China, NR IRB number 2021-28. Informed consent was obtained from all patients participating in the study.

In the same period, 19 normal subjects matched for age and sex were scanned as the control group. The inclusion criteria of the control group are age and gender matched healthy people without history and signs of nervous system diseases; The exclusion criteria were those with intracranial and extracranial metal implantation device, as well as imaging artifacts.

### 2.2. Clinical Assessment

Neurological impairment was assessed by using the National Institutes of Health Stroke Scale (NIHSS), with upper and lower extremity motor assessment as the basis of paresis grading (PG) [11]. The sum of upper and lower extremity motor scores ranged from 0 to 8 points. Assessments were made at the time of the first imaging examination and at 1 year after onset. According to the PG score at the last follow-up and the upper and lower extremity motor assessment scores, each patient was classified as having good motor outcomes (0–2) or poor motor outcomes (3–8), and as having good upper and lower extremity motor outcomes (0–1) or poor upper and lower motor outcomes (2–4). Brunel Balance Assessment (BBA) score, Modified Rankin Scale (mRS) score, and functional independence measure (FIM) motor items score were also obtained at the last follow-up.

### 2.3. MRI Protocol

All patient MRI scans (on a 1.5 Tesla GE magnetic resonance scanner (Signa HDx; GE Healthcare) obtained within 3 months after stroke onset. Diffusion tensor images (DTI) were obtained using single-shot spin-echo EPI sequence (repetition time: 10,000 ms; echo time: 115 ms; number of excitations: 2; matrix: 256 × 256; field of view: 256; pixels: 1.0 × 1.0 × 3.0 mm), with 80 standard axial images, and 15 noncollinear directions with a b-value of 1000 s/mm^2^. A total imaging time was less than 20 min.

### 2.4. Image Processing

DTI raw data were transferred to a separate workstation and analyzed by using Diffusion Toolkit 0.6.2.2 and TrackVis 0.6.0.1 (written by Ruopeng Wang, Martinos Center for Biomedical Imaging, Massachusetts General Hospital, Boston, MA, USA; https://www.trackvis.org; accessed on 18 September 2020) [12]. Dicom data were converted to the nifti format using the free dcm2nii software (http://www.mccauslandcenter.sc.edu/mricro/mricron/dcm2nii.html, accessed on 18 September 2020; output format FSL/SPM8-4D NIFTI). Pre-processing of DW images was performed with DTIPrep, which automatically corrects for eddy current distortions and head motion by removing low-quality directions and reorienting the b-matrix. Based on anatomy and T1-weighted imaging, regions of interest (ROIs) were manually drawn for each patient from two-dimensional (2D) axial fractional anisotropy (FA) color images of the ventral side of the bilateral CP and MCP (Figure 1) [13]. The FAs of the ROIs were automatically measured by the software.

Average FA and ADC values were calculated for CP-ROIs and MCP-ROIs. A FA ratio (rFA) was calculated by the FAs of the affected and that of unaffected sides using the following formula: CP rFA = CP affected side/CP unaffected side, and MCP rFA = MCP unaffected side/MCP affected side. Resulting values fall between 0 and 1. To compare the stroke-affected tracts with the unaffected tracts, a LI was calculated for each ROI/sequence comparison using the following formula: CP LI = (CP-FA affected side − CP-FA unaffected side)/(CP-FA affected side + CP-FA unaffected side), and MCP LI = (MCP-FA unaffected side—MCP-FA affected side)/(MCP-FA affected side + MCP-FA unaffected side).

### 2.5. Statistical Analysis

Statistical analyses were conducted by using SPSS software (version 22.o; IBM, Chicago, IL, USA). All data were expressed as means ± standard deviations. Group differences for continuous variables were tested using the Mann–Whitney test or Kruskal–Wallis test followed by post hoc Bonferroni correction, while categorical variables using the chi-square test or Fisher exact test as appropriate. Correlation analyses were performed using Pearson or Spearman correlation coefficients when appropriate. Receiver operating characteristic (ROC) curves were used to compare rFA and LI differences at different sites in predicting upper and lower extremity motor outcomes at 1 year after stroke onset, The value with a larger area under the curve was defined as the better predictive value. The Youden index was calculated to determine the cutoff point that classified the outcomes as good or poor. *p* values < 0.05 was considered statistically significantly.

## 3. Results

Of the 125 stroke patients, we excluded 58 patients and enrolled 67 patients. Consequently, we evaluated 44 patients (66%) in the outcome analysis, comprising 18 men and 26 women with a mean age of 58.9 ± 9.2 years (Figure 2).

Table 1 summarizes the clinical baseline characteristics of 44 patients. The proportion of stroke subtypes was 24 patients with ischemic stroke and 20 patients with hemorrhagic stroke. The total follow-up time ranged from 199 to 485 (359.8 ± 64.1) days. At the first follow-up, there were 35 cases with improved lower extremity motor function, 8 with no change, and 1 with aggravation; at the last follow-up, there were 38 cases with improved motor function, 5 with no change, and 1 with aggravation.

All patient images obtained within 3 months (mean, 44.0 days [±22.1] after stroke onset. The mean DTI parameters of the ventral CP and MCP and the comparisons between the normal control and case groups are shown in Table 2. Compared with the normal control group (right-handed), both the rFA and LI of the FA measured on the ventral side of the MCP and CP were significantly lower in the case group. Although the apparent diffusion coefficient ratios of the MCP and CP were also significantly lower in the case group, the LI in the case group was higher than that of the control group, with insufficient reliability.

Table 3 shows the relationships of the ventral rFA and LI of the MCP and CP with clinical outcomes within 3 months. The rFA (unaffected side/affected side) of the MCP within 3 months after onset was more significantly negatively correlated with 1-year PG, lower extremity PG, upper extremity PG, and NIHSS score (*p* = 0.000, r = −0.698; *p* = 0.000, r = −0.651; *p* = 0.000, r = −0.642; and *p* = 0.000, r = −0.519, respectively) and positively correlated with FIM motor items scores (*p* = 0.001, r = 0.487). The rFA of the CP was more significantly positively correlated with 1-year BBA score (*p* = 0.000, r = 0.581) and more significantly negatively correlated with mRS score (*p* = 0.001, r = −0.494). On the other hand, the LI of the CP was more significantly negatively correlated with 1-year upper extremity PG and mRS score (*p* = 0.000, r = −0.554 and *p* = 0.001, r = −0.49, respectively) and more significantly positively correlated with 1-year BBA score (*p* = 0.000, r = 0.573), compared with the LI of the MCP. The LI of the MCP was more significantly negatively correlated with 1-year PG, lower extremity PG, and NIHSS score (*p* = 0.000, r = −0.595; *p* = 0.000, r = −0.575; and *p* = 0.006, r = −0.407, respectively) and positively correlated with 1-year FIM motor items scores (*p* = 0.003, r = 0.443), compared with the LI of the CP.

Table 4 shows the relationships between different motor outcomes, upper and lower extremity motor outcomes at 1 year after stroke onset, and the variables examined within 3 months after stroke onset. No significant differences were noted in age and intraventricular hemorrhage for predicting total motor outcomes, or upper and lower extremity motor outcomes at 1 year after stroke onset. However, there was a significant difference between the lesions volume and total motor outcomes, as well as upper extremity motor outcomes (*p* = 0.017; *p* = 0.002, respectively), but it was not related to the lower extremity motor outcomes; there were significant differences between the severity of neurological deficit in the subacute stage and total motor outcomes, upper and lower extremity motor outcomes (*p* = 0.000; *p*= 0.016; and *p*= 0.032, respectively). Moreover, there were significant differences between the rFA and LI of the CP and MCP in the subacute phase and all motor outcomes at 1 year after stroke onset (both *p* = 0.000). Among them, in terms of the lower extremity motor outcome at 1 year, the rFA and LI values of MCP were the greater significantly different than those of CP (*p* = 0.000 vs. *p* = 0.003 and *p* = 0.000 vs. *p* = 0.002, respectively); In terms of the upper extremity motor outcomes at 1 year, the result was the opposite (*p* = 0.002 vs. *p* = 0.000 and *p* = 0.002 vs. *p* = 0.000, respectively).

Figure 3 and Figure 4 show the ROCs of the rFA and LI of the MCP and CP within 3 months for predicting lower and upper extremity motor outcomes at 1 year after stroke onset. For predicting lower extremity motor outcomes, the area under the curve of the rFA of the MCP, rFA of the CP, LI of the MCP, and LI of the CP were 0.893, 0.737, 0.866, and 0.741, respectively; the cutoff point for the rFA of the MCP and rFA of the CP were set at 0.925 (sensitivity: 79%, specificity: 100%) and 0.745 (sensitivity: 79%, specificity: 100%), respectively. For predicting upper extremity motor outcomes, the area under the curve of the rFA of the CP and rFA of the MCP were 0.86 and 0.755, respectively; the cutoff point for the rFA of the CP and rFA of the MCP were set at 0.775 (sensitivity: 100%, specificity: 72%) and 0.925 (sensitivity: 79%, specificity: 72%), respectively.

## 4. Discussion

This study found that within 3 months of stroke, the FA value of MCP distant lesions decreased by 10% on the unaffected side of the MCP compared with the affected side, consistent with CCD. Furthermore, FA decreased by 20% on the affected side of the CP compared with the unaffected side. At one year after stroke, the secondary degeneration showed significant negative correlations with hemiplegic motor outcomes and upper and lower extremity motor outcomes, as well as with the mRS score and FIM motor items score. Analysis of the differences in the recovery of upper and lower extremity motor ability indicated that the MCP was more closely related to lower extremity motor outcomes than CP (*p* = 0.000 and *p* = 0.003, respectively), whereas CP was more closely related to upper extremity motor outcomes than MCP (*p* = 0.000 and *p* = 0.002, respectively). Cutoff points for the rFAs of the MCP and CP were determined to be ≥0.925 and ≥0.775, respectively. ROC analysis confirmed that the area under the curve for MCP (0.893), was better than that of CP (0.737) for predicting lower extremity motor function, while the area under the curve of the CP (0.86) was slightly better than that of MCP (0.755) for predicting upper extremity motor function. We conclude that in patients with supratentorial stroke, the occurrence of CCD in the MCP and its severity within 3 months of onset (as measured by DTI) can predict lower extremity motor function recovery at one year after stroke.

The neural mechanism by which the recovery of walking in patients with hemiplegia after stroke is earlier and better than that of upper limb hand function is still unclear. Most of the published brain imaging studies have focused on the relationship between gray matter structure and walking ability [3,14]. For example, the prefrontal lobe in the cerebral cortex is the key area of walking [15], while the basal ganglia cerebellum and the pedunclopontine nucleus of reticular activation system is the key area for walking in the subcortical structure [16]. With regard to white matter damage, it is thought that white matter integrity, small vessel lesions (subcortical infarction and micro-bleeding) and functional connection changes are associated with the recovery of lower limb motion after stroke [14,17,18]. CST is one of the most important fiber bundles in terms of white matter integrity. With regard to direct injury, the current consensus is that CST (which mainly refers to the posterior limb of the internal capsule) injury is more closely related to motor outcomes and mid- or long-term prognosis than is the size of the supratentorial primary lesion [19,20]. In terms of the distant effect of lesions, the decreased FA value of CST at the CP level of ipsilateral lesions is negatively correlated with the overall prognosis [21,22]. In addition to CST, the integrity of the corpus callosum connecting both cerebral hemisphere is also related to the walking speed and gait after stroke [14]. Changes in the functional connection between bilateral central sulcus were also identified using functional brain imaging.

The MCP is the largest afferent pathway connecting the brainstem with the cerebellar hemisphere. It mostly receives information from each cerebral lobe cortex, crosses to the opposite side and enters the cerebellum after pontine substitution. This anatomical structure is consistent with the CCD phenomenon observed secondary to supratentorial lesions of various etiologies [1,23,24,25,26,27]. However, there has been little discussion of MCP in the study of white matter integrity in stroke [28,29]. The present data confirmed that the integrity of the CST fiber bundle on the ventral side of CP was more closely associated with the outcome of upper limb motor function than the integrity of the MCP fiber bundle consistent with the findings of most CST Studies. On the other hand, the MCP fiber bundle may play a more important role in the outcome of lower limb motor function. A recently published study reported that stimulation of the contralateral cerebellar hemisphere by intermittent theta brain simulation (ITBs) after stroke appeared to improve gait and balance more significantly than stimulation of the M1 area [30]. MCP is an important node of cortical basal-ganglia–cerebellar–thalamus circuit, it is therefore reasonable to speculate that MCP plays a more important role in CCD phenomenon and in the gait damage recovery mechanism after stroke than does CST.

Little is known about the neural mechanisms involved in the recovery of walking ability after stroke [4]. An important reason for this is that human beings are the only mammals that walk upright. Hence, animal models cannot replace human beings in studies of locomotion involving the two lower limbs. In the early stage of hemiplegia, lower limb motor function is mostly evaluated by muscle strength due to the inability to walk [13,30,31]. Moreover, with partial recovery of lower limb mobility in the early stage of rehabilitation, it can be difficult to objectively record the dynamic changes in hemiplegic gait. This is due to the instability of motion trajectory and torque data in clinical work, with the exception of spatiotemporal parameters. Therefore, it is more feasible to record the activity of lower limb muscle groups with electrophysiology. Using magnetic stimulation motor-evoked potential, Jayaram et al. [32] recorded the electrophysiological changes in the proximal muscles of the lower limbs on the hemiplegic side and non-hemiplegic sides of patients with chronic stroke. They also observed the structural changes in the brain using DTI. These workers found that the greater the ipsilateral functional connection between the contralateral motor cortex of the lesion and the lower limbs on the paralyzed side (measured by motor evoked potential), the worse the walking ability of the patient, and the more obvious the structural damage to CST in the hemisphere of the lesion. However, Jayaram et al. [32] did not report whether this phenomenon occurred synchronously with CCD, and did not record EEG changes in the corresponding brain area before and after stimulation. Therefore, the synchronous connection between EMG signal, which may be the secondary signal after denervation, and the contralateral brain area of the lesion, which is also on this side if CCD occurs, requires further investigation.

This present study has some limitations. First, the sample size was quite small and hence, the conclusion needs to be confirmed by larger studies. Secondly, the region of interest (ROI) selection method used in this study is one of the common FA value methods, but some bias because of the manual selection is inevitable. The density weighted template [33] and 3-D individual trajectory [34] methods may be more reliable DTI measurement methods. With the increasingly urgent demand for closed-loop neuromodulation technology based on the brain-computer interface and artificial intelligence, identifying the target at the millisecond level based on DTI and other imaging technologies poses a new challenges for the clinical application of DTI technology [22]. Finally, the mechanism linking secondary impairment of the MCP with lower extremity motor recovery after stroke, including the CCD mechanism, needs further study. This is especially important at the neural circuit and molecular levels, as its occurrence is not isolated from the whole brain [29,35].

## 5. Conclusions

In this study, the results support that MCP secondary degeneration is common within 3 months after supratentorial stroke, which is consistent with CCD; This remote lesion of MCP may be more closely related to gait recovery after hemiplegia than upper limb recovery, and its rFA value can be used clinically to predict the outcome of lower limb movement.

## Figures and Tables

**Figure 1 brainsci-13-00412-f001:**
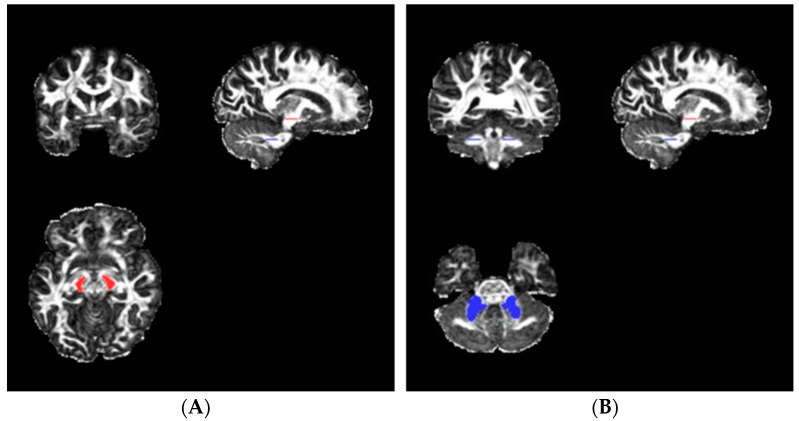
Region of interest definition. (**A**) Cerebral peduncle ROI. (**B**) Middle cerebellar peduncle ROI.

**Figure 2 brainsci-13-00412-f002:**
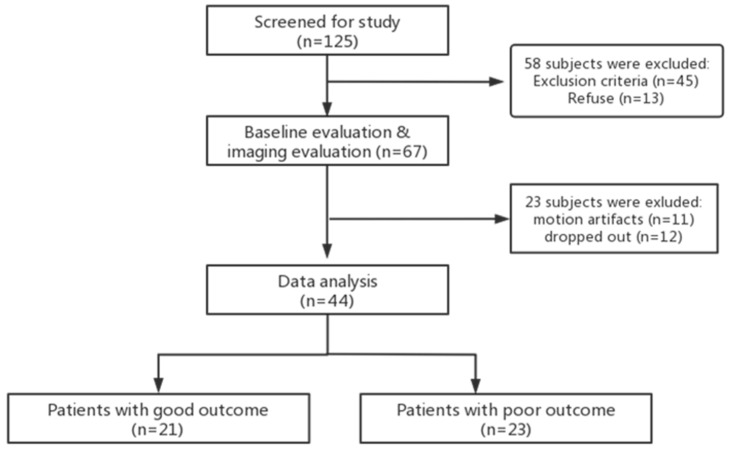
Flowchart for patient inclusion and exclusion.

**Figure 3 brainsci-13-00412-f003:**
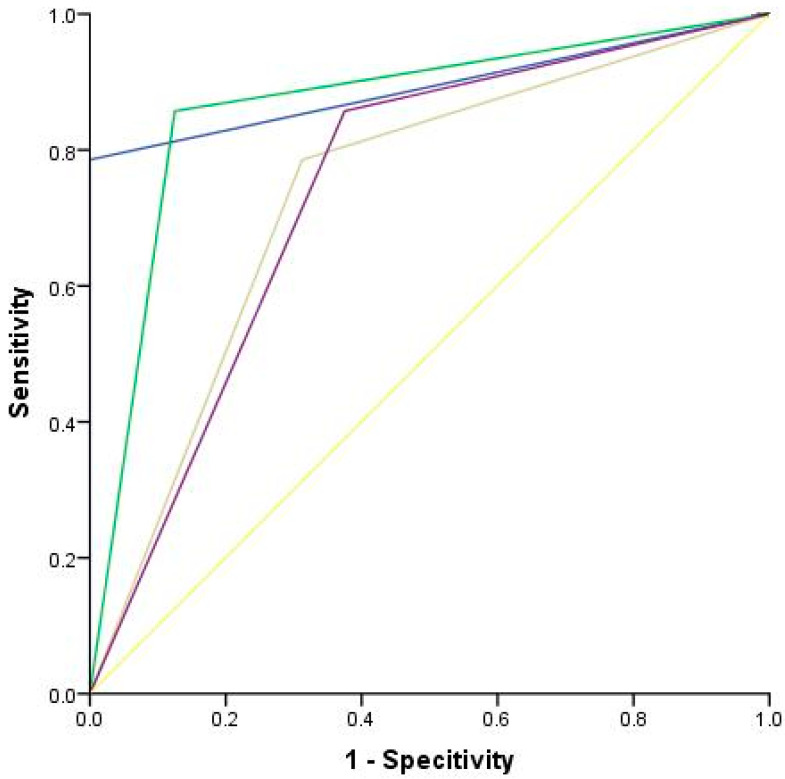
ROC curves for rFA and FA Laterality Index within 3 months post-stroke for prediction of lower-Extremity motor outcome by LE-PG ≤ 1 after 1 year. The areas under the curve were both higher for rFA (blue line) and FA laterality index (green line) obtained in MCP than that in CP (0.893 vs. 0.737, 0.866 vs. 0.741, respectively). The purple and crown line represent rFA and FA laterality index obtained in CP respectively. The straight light-yellow line represents 50% of the area under the curve.

**Figure 4 brainsci-13-00412-f004:**
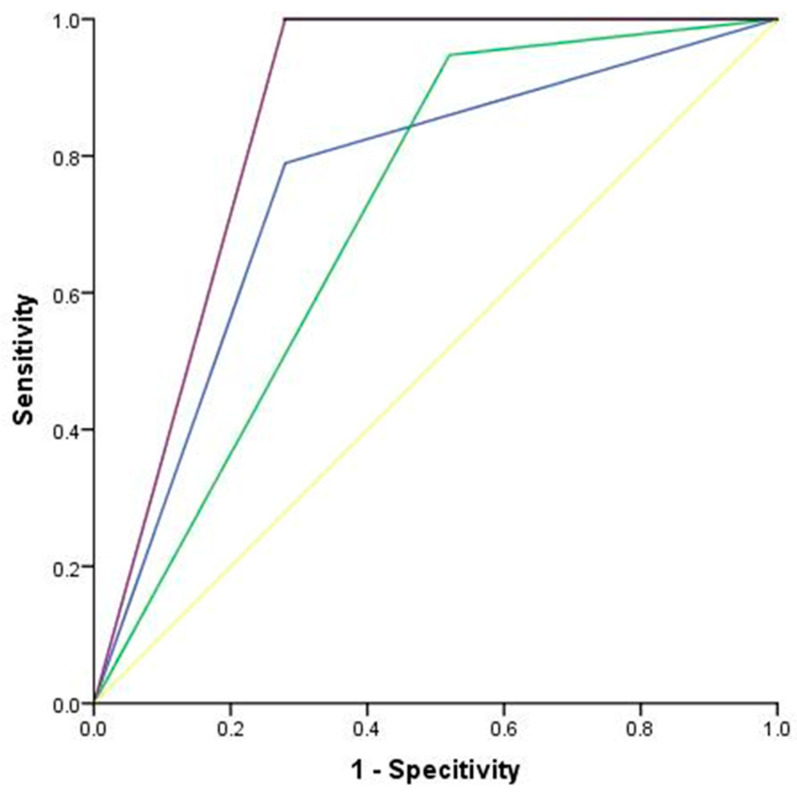
ROC curves for rFA and FA Laterality Index within 3 months post-stroke for prediction of Upper-Extremity motor outcome by UE-PG ≤ 1 after 1 year. The areas under the curve were lower for rFA (blue line) and FA laterality index (green line) obtained in MCP than that in CP (0.755 vs. 0.86, 0.714 vs. 0.86, respectively). The purple line represents rFA obtained in CP. The straight light-yellow line represents 50% of the area under the curve.

**Table 1 brainsci-13-00412-t001:** Patients’ Demographical and Clinical Data Characteristics.

	All Patients	MCA Infarction	ICH	Control
# of patients	44	24	20	19
Age (Yrs)	58.9 ± 9.2	58.5 ± 10.2	59.5 ± 8.0	55.95 ± 7.37
Gender (%, Female)	18	21	15	42
Lesion side (%, Right)	31.8	33.3	30	/
Lesion Volume (in cc)	36.5 ± 38.2	45.4 ± 46.7	25.9± 20.9	/
Handedness (%, Right)	100	100	100	/
NIHSS, baseline	12.1 ± 4.6	11.1 ± 5.0	13.2 ± 3.8	/
NIHSS, 12-months	7.0 ± 4.6	6.8 ± 4.7	7.3 ± 4.6	/
UE-PG, baseline(%, ≤1)	2.3	4.2	0	/
UE-PG, 12-months(%, ≤1)	43.2	41.7	45	/
LE-PG, baseline(%, ≤1)	25	25	25	/
LE-PG, 12-months(%, ≤1)	63.6	66.7	60	/
BBA, 12-months	8.6 ± 3.5	9.6 ± 2.9	7.5 ± 3.9	/
FIM motor sub-score, 12-months	70.3 ± 20.4	73.6 ± 18.8	66.5 ± 22.0	/
MRS, 12-months (%, ≤2)	31.8	41.7	20	/
Imaging days post-stroke (days)	44.0 ± 22.1	42.6 ± 17.3	45.7 ± 27.2	/
physical therapy duration (days)	84.5 ± 71.2	82.2 ± 67.6	87.3 ± 77.0	/
Following up days post-stroke(days)	359.8 ± 64.1	351.5 ± 45.4	369.8 ± 81.4	/
Hypertension (%)	55	46	65	5
Hyperlipidemia (%)	52	58	45	16
Diabetes (%)	36	46	25	5
Coronary artery disease (%)	32	36	28	21
Atrial Fibrillation (%)	25	29	20	0
Smoking (%)	50	50	50	32
Alcohol (%)	48	46	50	26

Note: UE-PG = Upper-Extremity Paresis Grading; LE-PG = Lower-Extremity Paresis Grading; NIHSS = National Institutes of Health Stroke Scale; BBA = Brunel Balance Assessment; FIM motor sub-score = Motor subscale of the Functional Independence Measure; MRS = Modified Rankin Scale; All values are means and standard deviation (in brackets).

**Table 2 brainsci-13-00412-t002:** DTI parameter in controls and patients.

	Controls ^a^	All Patients ^b^	ICH ^b^	IS-MCA ^b^
CP				
rFA	0.946 ± 0.021	0.794 ± 0.015 *	0.745 ± 0.142 *	0.835 ± 0.138 *
FA LI	0.008 ± 0.03	−0.123 ± 0.092 *	−0.156 ± 0.092 *	−0.096 ± 0.084 *
rADC	0.951 ± 0.021	0.877 ± 0.099 *	0.86 ± 0.098 *	0.892 ± 0.1
ADC LI	0.002 ± 0.03	0.019 ± 0.089	0.021 ± 0.098	0.017 ± 0.083
MCP				
rFA	0.972 ± 0.021	0.909 ± 0.0138 *	0.927 ± 0.145	0.894 ± 0.132 *
FA LI	−0.01 ± 0.085	−0.046 ± 0.082 *	−0.028 ± 0.087 *	−0.061 ± 0.075 *
rADC	0.951 ± 0.032	0.905 ± 0.068 *	0.92 ± 0.054	0.892 ± 0.076 *
ADC LI	−0.012 ± 0.029	0.022 ± 0.06 *	0.029 ± 0.044 *	0.016 ± 0.072

Note: ICH: intracerebral hemorrhage; IS-MCA: ischemic stroke in the territory of the middle cerebral artery; CP = Cerebral Peduncle; MCP = Middle Cerebellar Peduncle; rFA = rate of the Fractional Anisotropy; LI = Laterality Index; rADC = rate of the Apparent Diffusion Coefficient; the formulas a as follow: CP rFA calculated using formula (FAL/FAR); MCP rFA calculated using formula (FAR/FAL);CP Laterality Index calculated using formula (FAL − FAR)/(FAL + FAR); MCP Laterality Index calculated using formula (FAR − FAL)/(FAL + FAR); the formulas b as follow: CP rFA calculated using formula (FAA/FAU); MCP rFA calculated using formula (FAU/FAA); CP Laterality Index calculated using formula (FAA − FAU)/(FAA + FAU); MCP Laterality Index calculated using formula (FAU − FAA)/(FAA + FAU); * Significantly different from zero with *p* < 0.05.

**Table 3 brainsci-13-00412-t003:** Statistical analysis comparing the FA ratio and Laterality Index obtained at CP and MCP as well as the clinical scores.

	CP rFA		CP LI ^a^		MCP rFA		MCP LI ^b^	
	r *	*p* Value ^a^			r *	*p* Value		
NIHSS, 12-months	−0.403	0.007	−0.403	0.007	−0.519	0.000	−0.407	0.006
UE-PG, 12-months	−0.565	0.000	−0.554	0.000	−0.642	0.000	−0.528	0.000
LE-PG, 12-months	−0.386	0.010	−0.372	0.013	−0.651	0.000	−0.575	0.000
PG, 12-months	−0.541	0.000	−0.528	0.000	−0.698	0.000	−0.595	0.000
BBA	0.581	0.000	0.573	0.000	0.547	0.004	0.452	0.002
MRS	−0.494	0.001	−0.49	0.001	−0.430	0.004	−0.344	0.022
FIM motor sub-score	0.435	0.003	0.43	0.004	0.487	0.001	0.443	0.003

Note: CP = Cerebral Peduncle; MCP = Middle Cerebellar Peduncle; rFA = rate of the Fractional Anisotropy; LI ^a^ = Laterality Index calculated using formula (FAA − FAU)/(FAA + FAU); LI ^b^ = Laterality Index calculated using formula (FAU − FAA)/(FAA + FAU); * Spearman rank correlation coefficient was used for all comparisons.

**Table 4 brainsci-13-00412-t004:** Relationship between patient motor outcome and various parameters *.

	Motor Outcome (*n*)		Lower Extremity Motor Outcome (*n*)		Upper Extremity Motor Outcome (*n*)	
	Good	Poor	Good	Poor	Good	Poor
Age ≥ 65 years						
Yes	7	7	8	6	6	8
No	14	16	20	10	13	17
*p* value	1		0.738		1	
NIHSS ≥ 8						
Yes	10	22	17	15	10	22
No	11	1	11	1	9	3
*p* value	0.000 †		0.032 †		0.016 †	
Lesion volume ≥ 30 mL						
Yes	5	14	10	9	3	16
No	16	9	18	7	16	9
*p* value	0.017 †		0.220		0.002 †	
Intraventricular bleeding (for ICH)						
Yes	2	7	4	5	2	7
No	7	4	8	3	7	4
*p* value	0.092		0.362		0.092	
CP rFA ≥ 0.745 ^#^						
Yes	19	7	22	5	19	7
No	2	16	6	11	0	18
*p* value	0.000 †		0.003 †		0.000 †	
MCP rFA ≥ 0.925						
Yes	17	5	22	0	15	7
No	4	18	6	16	4	18
*p* value	0.000 †		0.000 †		0.002 †	
CP LI ≥ −0.16895 ^@^						
Yes	19	7	24	6	19	7
No	2	16	4	10	0	18
*p* value	0.000 †		0.002 †		0.000 †	
MCP LI ≥ −0.04975 ^$^						
Yes	17	6	24	2	18	13
No	4	17	4	14	1	12
*p* value	0.000 †		0.000 †		0.002 †	

* The Spearman rank correlation coefficient was used for all comparisons. # the cutoff point of the better predictive value for upper extremity motor outcome is 0.775; @ the cutoff point of the better predictive value for upper extremity motor outcome is −0.12698; $ the cutoff point of the better predictive value for upper extremity motor outcome is −0.07750. † Statistically significant.

## Data Availability

The data presented in this study are available on request from the corresponding author.

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
