# Peer review of "Long-Term Lower Limb Motor Function Correlates with Middle Cerebellar Peduncle Structural Integrity in Sub-Acute Stroke: A ROI-Based MRI Cohort Study"

_brainsci, 2023, doi:10.3390/brainsci13030412_

Round 1

Reviewer 1 Report

First of all, congregations for authors for this interesting and important work. Please find my following comment in order to quality-assure the manuscript:

1- The manuscript has a lot of abbreviations that make the reader confused

2- Please justify more the selection of middle cerebellar peduncle rather than superior or inferior cerebellar peduncle.

3- The objective of the study in the last of introduction, did the study assess the upper extremity status. The title is regarding only the lower limb.

4- Line 253-254: it needs a reference

5- The major concern regarding the study is discussion.
A- The discussion did not explain the results.
B- Lines 289-310: I think that is out of the context.

C- There is no clinical or research recommendations regarding the results.

6- Are the conclusions supported by the results?

Author Response

1-The manuscript has a lot of abbreviations that make the reader confused
Reply: Some abbreviations are adjusted or deleted for readability. Such as MCA.

2- Please justify more the selection of middle cerebellar peduncle rather than superior or inferior cerebellar peduncle.

Reply: Among the complex tracts connections between the brain and cerebellum, the middle cerebellar peduncle (MCP) is the largest fibrous structure, which only contains afferent nerve bundles, namely, pontocerebellar tract (PT). Most of the fiber bundles from the cerebral cortex intersect and enter the cerebellum through the middle cerebellar peduncle. Functionally, this structure is the continuation of the projection of the cortical cerebellum to the pons after synaptic nerve conversion [1]. Since CCD is secondary to brain lesions, we believe that MCP damage can be equivalent to CCD to some extent.

 [1] Duus: Topical Diagnosis in Neurology[M], 4th Ed, Georg Thieme Verlag, 2005.

3- The objective of the study in the last of introduction, did the study assess the upper extremity status. The title is regarding only the lower limb.

 Reply: This paper aims to highlight the indispensable role that MCP may play in lower limb motion control. Therefore, we think it is necessary to use upper limb motion related data as a reference in the introduction.

  • Line 253-254: it needs a reference

Reply: References have been marked to the end of the sentence.

  • The major concern regarding the study is discussion.
  • The discussion did not explain the results.
  • Lines 289-310: I think that is out of the context.
  • There is no clinical or research recommendations regarding the results.
  • the A-discussion section on Lines 234-251 further elaborated the data presented in the results. According to the statistical differences in the data, it was suggested that MCPbeing far from the primary lesions had secondary degeneration within 3 months of stroke. The degree of MCP degeneration, compared with that of CP, was more closely related to the poor long outcome of lower limb movement 1 year later.

  • Line 289-310 aims to further explain the unique neural mechanism of hemiplegic gait recovery, but it is unique for humans to walk upright, so the data from animal experiments cannot explain this mechanism. At present, only in vivo electrophysiological research (such as MEP, TMS-EEG) combined with structural and functional imaging or the next feasible research scheme is available.

C - Research suggestions are the same as B

 6- Are the conclusions supported by the results?

Reply: It may be more reasonable to modify the statement in the conclusion as follows:

The results support that MCP secondary degeneration is common within 3 months after supratentorial stroke, which is consistent with CCD; This remote lesion of MCP may be more closely related to gait recovery after hemiplegia than upper limb recovery, and its rFA value can be used clinically to predict the outcome of lower limb movement.

Reviewer 2 Report

Summary

Wang and colleagues investigated the relationship between DTI derived metrics and clinical scores of motor performance in patients with stroke. They reported moderate to low correlations between imaging metrics and clinical measures obtained 12 months after stroke onset. They concluded that the middle cerebellar peduncle could play a significant role in recovery of walking ability after stroke as DTI metrics obtained at its level showed to be more related to motor performance compared to cerebral peduncle DTI metrics.  

Major comments

-     The Manuscript lacks some relevant references. For instance, in the Introduction section, references about the possibility of assessing CCD with diffusion tensor imaging are missing, as well as references about cerebral and cerebellar control on motor functions.

-        Did the Authors confirm the presence of CCD with PET or CT/MRI perfusion?

-        The characteristics of the control group are missing. Are the controls comparable to the patients in terms of demographic characteristics?

-       Did the Authors perform all the preprocessing steps such as motion correction and Eddy current correction before DTI metrics extraction?

-       The Authors stated that the clinical scores were assessed 12 months after stroke onset, but they also declared that the “total follow-up time ranged from 199 to 485 (359.8 ± 64.1) day”, implying that the follow-up time was highly variable and, for some patients, shorter than a year. This might represent a methodological issue since the Authors did not account for the variability in follow-up time in the statistical analysis.

-       The Discussion section does not properly follow the findings of the Manuscript, as the Authors digressed on corticospinal tract (which has not been directly evaluated in the present study), on Parkinson’s disease (which presents a completely different etiopathogenesis compared to stroke), transcranial magnetic stimulation and electrophysiology.

Additional comments

-       The paper could benefit from a native English speaker revision, as quite some grammar errors can be observed, and many sentences are hard to understand.

-        Exclusion and inclusion criteria for the control group are missing.

-        The Authors should clarify how they evaluated lesions volume.

Author Response

   The Manuscript lacks some relevant references. For instance, in the Introduction section, references about the possibility of assessing CCD with diffusion tensor imaging are missing, as well as references about cerebral and cerebellar control on motor functions.

Reply: Relevant literature has been supplemented or added and cited in the introduction.

[1] Liu, G., Guo, Y., Dang, C., Peng, K., Tan, S., Xie, C., Xing, S., & Zeng, J. (2021). Longitudinal changes in the inferior cerebellar peduncle and lower limb motor recovery following subcortical infarction. BMC neurology, 21(1), 320. https://doi.org/10.1186/s12883-021-02346-x

[2] Soulard, J., Huber, C., Baillieul, S., Thuriot, A., Renard, F., Aubert Broche, B., Krainik, A., Vuillerme, N., Jaillard, A., & ISIS-HERMES Group (2020). Motor tract integrity predicts walking recovery: A diffusion MRI study in subacute stroke. Neurology, 94(6), e583–e593. https://doi.org/10.1212/WNL.0000000000008755

-        Did the Authors confirm the presence of CCD with PET or CT/MRI perfusion?

Reply: Due to the addition of PET or perfusion imaging evaluation in clinical practice, there are ethical limitations; Due to the limitation of imaging equipment in a hospital where the author's team belongs, it is regrettable that the CCD and MCP changes cannot be evaluated simultaneously. This has a certain impact on the results of the article.

-        The characteristics of the control group are missing. Are the controls comparable to the patients in terms of demographic characteristics?

Reply: The demographic characteristics of the control group are supplemented and shown in Table 1.

All Persons

# of Nomal Persons

19

Age (Yrs)

55.95±7.37

Gender (%, Female)

42

Hypertension (%)

5

Hyperlipidemia (%)

16

Diabetes (%)

5

Coronary artery disease (%)

21

Atrial Fibrillation (%)

0

Smoking (%)

32

Alcohol (%)

26

-       Did the Authors perform all the preprocessing steps such as motion correction and Eddy current correction before DTI metrics extraction?

Reply: 

Dicom data were converted to the nifti format using the free dcm2nii software (http://www.mccauslandcenter.sc.edu/mricro/mricron/dcm2nii.html, output format FSL/SPM8—4D NIFTI).

Pre-processing of DW images was performed with DTIPrep, which automatically corrects for eddy current distortions and head motion by removing low-quality directions and reorienting the b-matrix.[1]

[1] Oguz I, Farzinfar M, Matsui J, Budin F, Liu Z, Gerig G, Johnson HJ, Styner M (2014) DTIPrep: quality control of diffusion weighted images. Front Neuroinform 8:4

-       The Authors stated that the clinical scores were assessed 12 months after stroke onset, but they also declared that the “total follow-up time ranged from 199 to 485 (359.8 ± 64.1) day”, implying that the follow-up time was highly variable and, for some patients, shorter than a year. This might represent a methodological issue since the Authors did not account for the variability in follow-up time in the statistical analysis.

Reply: The possibility of data bias caused by the impact of different follow-up time on the outcome exists, but this kind of situation is mainly seen in different surgical methods of tumor and surgery. In clinical practice, the difference of function change with time within 0.5-2 years after stroke is significantly smaller than that within 1-6 months after stroke, which is the so-called platform stage. Although the change of brain remodeling may still have significant changes at this time. This can be reflected in many literatures on the prognosis of stroke. Therefore, time scale is seldom used as a dependent variable in clinical research of stroke.

-       The Discussion section does not properly follow the findings of the Manuscript, as the Authors digressed on corticospinal tract (which has not been directly evaluated in the present study), on Parkinson’s disease (which presents a completely different etiopathogenesis compared to stroke), transcranial magnetic stimulation and electrophysiology.

Reply: The discussion part has been rewritten to clarify the views supported by the data presented in this paper; Assessment of corticospinal tract This study was expressed by measuring FA value of cerebral peduncle (midbrain); For the avoidance of ambiguity, the text on gait cerebellar stimulation therapy for Parkinson's disease is adjusted to the evidence that cerebellar stimulation participates in the reconstruction of brain-cerebellar motor network after hemiplegia; This paper discusses the therapeutic intervention targeted to cerebellum and electrophysiological evaluation, aiming to explore the potential neural mechanism of MCP secondary degeneration by measuring cortical synaptic potential in vivo in the future.

[1] Koch, G., Bonnì, S., Casula, E. P., Iosa, M., Paolucci, S., Pellicciari, M. C., Cinnera, A. M., Ponzo, V., Maiella, M., Picazio, S., Sallustio, F., & Caltagirone, C. (2019). Effect of Cerebellar Stimulation on Gait and Balance Recovery in Patients With Hemiparetic Stroke: A Randomized Clinical Trial. JAMA neurology, 76(2), 170–178. https://doi.org/10.1001/jamaneurol.2018.3639

Additional comments

-       The paper could benefit from a native English speaker revision, as quite some grammar errors can be observed, and many sentences are hard to understand.

Reply: According to the editor's suggestion, Native writer with medical background have been invited to modify the text of the introduction and discussion.

-        Exclusion and inclusion criteria for the control group are missing.

Reply: The inclusion criteria of the control group are age and gender matched healthy people without history and signs of nervous system diseases; The exclusion criteria were those with intracranial and extracranial metal implantation device , as well as imaging artifacts.

-        The Authors should clarify how they evaluated lesions volume.

Reply: Volume=sum of all layer areas of abnormal area × (layer thickness+layer spacing)

Reviewer 3 Report

An article with great potential. Some issues require clarification.

1. The authors analyze rFA, LI, rADC. FA or ADC derived parameters (here the question is whether we have MD - Mean difusivisty, the mean of the eigenvalues of the tensor after diagonalization, or ADC, calculated from 3 orthogonal DWIs, theoretically it's almost the same, but in practice, not). Why ?

I would expect to look for differences (statistically significant) on the base parameters. It would be nice to see the original values, FA, MD and their standard deviations, SD.

2. The authors do not take into account the potential influence of systematic errors related to the non-uniformity of the magnetic field gradients. Potential impacts on results should be discussed. The BSD-DTI type problem shows the possible extent of systematic errors and the impact on DTI parameters. This is important because the data comes from a single MR scanner.

Author Response

  1. The authors analyze rFA, LI, rADC. FA or ADC derived parameters (here the question is whether we have MD - Mean difusivisty, the mean of the eigenvalues of the tensor after diagonalization, or ADC, calculated from 3 orthogonal DWIs, theoretically it's almost the same, but in practice, not). Why ?

I would expect to look for differences (statistically significant) on the base parameters. It would be nice to see the original values, FA, MD and their standard deviations, SD.

Reply: There are differences between FA/MD/ADC values, and FA value is relatively better in assessing white matter fiber integrity; If we consider the effect of acute cellular edema or perfusion changes on brain tissue, MD or ADC may be more appropriate, The FA value used in this paper is more consistent with the research purpose. The following table shows the average and standard deviation of FA, MD, ADC in the four ROI regions of the original cerebral hemorrhage, cerebral infarction and normal control group for reference.

Region-of-interest analysis in the patients with stroke (mean±SD)

Affected side

Unaffected side

Ratio†

Group comparison

 (p value)‡

CP FA (dimensionless)

0.384±0.065

0.488±0.052

0.79

0.000§

MCP FA

0.471±0.062

0.521±0.051

0.91

0.001§

CP MD(10–3mm2/s)

0.346±0.081

0.354±0.079

0.98

0.262

MCP MD

0.430±0.111

0.419±0.113

0.97

0.148

CP ADC(10–3mm2/s)

0.976±0.137

0.938±0.118

1.04

0.171

MCP ADC

0.899±0.116

0.939±0.121

0.96

0.024§

 CP= Cerebral Peduncle; MCP= Middle Cerebellar Peduncle

† Affected side/unaffected side. ‡ Wilcoxon rank-sum test. § Statistically significant

  1. The authors do not take into account the potential influence of systematic errors related to the non-uniformity of the magnetic field gradients. Potential impacts on results should be discussed. The BSD-DTI type problem shows the possible extent of systematic errors and the impact on DTI parameters. This is important because the data comes from a single MR scanner.

Reply: There are systematic errors in the magnetic field gradient of different types of MR scanners, which are difficult to avoid in clinical research. Moreover, because the research time span of the clinical queue is long, even the same machine has errors in the time dimension. A theoretically feasible way is to perform shimming before each examination, but this is also difficult to achieve in the clinical environment. The direct result caused by the inhomogeneity of magnetic field gradient is the absolute value bias of DTI parameter measurement. One way to solve this problem is to use the relative value, which is also the reason why the ratio is used in this paper.

Round 2

Reviewer 1 Report

The authors well addressed the comments 

Reviewer 2 Report

I thank the Authors for adding all the requested information and having their Manuscript revised.

I believe that the Manuscript has been ameliorated and could be published in its current status.